# The Role of Transcranial Magnetic Stimulation, Peripheral Electrotherapy, and Neurophysiology Tests for Managing Incomplete Spinal Cord Injury

**DOI:** 10.3390/biomedicines11041035

**Published:** 2023-03-27

**Authors:** Katarzyna Leszczyńska, Juliusz Huber

**Affiliations:** 1Department of Pathophysiology of Locomotor Organs, Poznan University of Medical Sciences, 28 Czerwca 1956 no 135/147, 60-545 Poznań, Poland; kleszczynska@ump.edu.pl; 2Department of Neurosurgery, Wroclaw Medical University, Borowska 213, 50-556 Wroclaw, Poland

**Keywords:** incomplete spinal cord injury, neuromodulation methods, nerve electrostimulation, repetitive transcranial magnetic stimulation, kinesiotherapy, electromyography

## Abstract

Efforts to find therapeutic methods that support spinal cord functional regeneration continue to be desirable. Natural recovery is limited, so high hopes are being placed on neuromodulation methods which promote neuroplasticity, such as repetitive transcranial magnetic stimulation (rTMS) and electrical stimulation used as treatment options for managing incomplete spinal cord injury (iSCI) apart from kinesiotherapy. However, there is still no agreement on the methodology and algorithms for treatment with these methods. The search for effective therapy is also hampered by the use of different, often subjective in nature, evaluation methods and difficulties in assessing the actual results of the therapy versus the phenomenon of spontaneous spinal cord regeneration. In this study, an analysis was performed on the database of five trials, and the cumulative data are presented. Participants (iSCI patients) were divided into five groups on the basis of the treatment they had received: rTMS and kinesiotherapy (N = 36), peripheral electrotherapy and kinesiotherapy (N = 65), kinesiotherapy alone (N = 55), rTMS only (N = 34), and peripheral electrotherapy mainly (N = 53). We present changes in amplitudes and frequencies of the motor units’ action potentials recorded by surface electromyography (sEMG) from the tibialis anterior—the index muscle for the lower extremity and the percentage of improvement in sEMG results before and after the applied therapies. The increase in values in sEMG parameters represents the better ability of motor units to recruit and, thus, improvement of neural efferent transmission. Our results indicate that peripheral electrotherapy provides a higher percentage of neurophysiological improvement than rTMS; however, the use of any of these additional stimulation methods (rTMS or peripheral electrotherapy) provided better results than the use of kinesiotherapy alone. The best improvement of tibialis anterior motor units’ activity in iSCI patients provided the application of electrotherapy conjoined with kinesiotherapy and rTMS conjoined with kinesiotherapy. We also undertook a review of the current literature to identify and summarise available works which address the use of rTMS or peripheral electrotherapy as neuromodulation treatment options in patients after iSCI. Our goal is to encourage other clinicians to implement both types of stimulation into the neurorehabilitation program for subjects after iSCI and evaluate their effectiveness with neurophysiological tests such as sEMG so further results and algorithms can be compared across studies. Facilitating the motor rehabilitation process by combining two rehabilitation procedures together was confirmed.

## 1. Introduction

Incomplete spinal cord injury (iSCI) handicaps neuronal impulse transmission between the supraspinal centres and muscles, leading to varying degrees of paralysis and sensory impairment that greatly impact functional ability. According to the data presented by the WHO, between 250,000 and 500,000 people suffer a spinal cord injury (SCI) every year, and millions of people are currently living with this highly disabling injury [1]. Speaking of causes of SCI, car accidents rank first, with a score of 37.5%, followed by falls with a score of 31.3% in the U.S. population. Thanks to medical advances, the length of survival after spinal cord injuries has been significantly extended [2]. These are people living with disabilities that affect not only them but their entire environment and are very costly to treat and severely hampered in their daily lives. Many of these people do not return to work and are sentenced to the lifetime costs of rehabilitation and long-term care expenses [2]. Approximately 50% of subjects affected by spinal cord injury present a lesion in the cervical and thoracic regions, so there is a high percentage of this population that present severe limitations in the execution of basic activities of daily life [3]. Regardless of the mechanism of injury and highly heterogeneous pathophysiology [4], most often, patients have varying degrees of loss to the sensory and motor functions below the level of the injury, the presence of pain, and a wide range of autonomic dysfunctions. In addition, people with iSCI have many problems with mental health, and their condition very often affects the entire community [5]. Over the past decades, neurosurgical and rehabilitation advances have opened new horizons for spinal cord regeneration. Today, efforts to find methods that support cortical and spinal cord plasticity and functional recovery after iSCI continue to be even more desirable and still remain controversial [6]. Although many different treatment options have been suggested for the management of SCI, including surgery, kinesiotherapy, physical therapy, cell-based therapy, or a combination of these approaches [5,6,7,8,9], there is still no agreement on which method provides the best chance of clinical improvement [10]. Moreover, the more presented clinical interventions are often difficult to compare due to the lack of standardised objective evaluation methods.

Natural, spontaneous spinal cord recovery, which occurs in most individuals with SCI, is limited [7,10], so high hopes are being placed on neuromodulation and stimulation methods such as repetitive transcranial magnetic stimulation (rTMS) and peripheral electrical stimulation used as treatment options for managing complete and incomplete spinal cord injury (iSCI) [10,11,12,13,14]. Thus, we have focused on providing evidence of two treatment options’ effectiveness that may have the potential to support and enhance functional recovery in patients with iSCI. In so doing, we have conducted studies on repetitive transcranial magnetic stimulation (rTMS) combined with kinesiotherapy [15], as well as peripheral nerve electrostimulation combined with kinesiotherapy [16]. Both methods provided better results in clinical neurophysiology tests compared to the group treated with kinesiotherapy alone. In this study, another analysis was performed in order to compare these two neuromodulation methods and kinesiotherapy. Our goal was to assess which of these treatment options provides the best chance of clinical improvement for iSCI patients. We also reviewed the current most frequently applied protocols of rTMS and peripheral electrotherapy and presented a review of the available in-literature algorithms regarding both methods used for the management of iSCI patients. Moreover, we want to provide more evidence concerning the application of both types of stimulations in the spinal cord injury rehabilitation process. As proof steadily grows in favour of electrical and magnetic stimulation as therapeutic tools, there is a need to develop standardised algorithm protocols for their application and evaluation. It is our hope and goal to provide the foundation and motivation for other clinicians to investigate and implement both types of stimulation and assess their effectiveness with neurophysiological tests so further results can be compared across studies. This likely will lead to evidence-based neuromodulation and neurorehabilitation programs for people after incomplete spinal cord injury [17] and will, as a result, improve the quality of life of these patients.

## 2. Materials and Methods

For the purposes of this work, we combined and upgraded the data from two previous studies, created a new set of data, and added results of the treatment in two groups of iSCI patients who received either rTMS or electrotherapy treatment, in order to try to answer the question of whether one of the neuromodulation and treatment methods is significantly better. We upgraded about 30% of the data in each patient group. We chose the data for patients by considering the spinal injury levels, similar ASIA C and D incidences in each group, averaged time from injury, and average before-after treatment observation time, as well as the BMI characteristics and gender. In addition, we once again gathered and recalculated the tibialis anterior muscle surface electromyography (sEMG) recording results for patients treated with kinesiotherapy alone, and this time, we presented the results for a total group of 55 patients.

### 2.1. Patients and Healthy Subjects Included in the Study

The study on the effects of peripheral electrostimulation only (Electro group) included 53 patients, while peripheral electrostimulation conjoined with kinesiotherapy (Electro + K group) included 65 patients with confirmed C4 to Th12 spinal cord injury. The study on the application of rTMS only (rTMS group) included 34 patients, and rTMS conjoined with kinesiotherapy (rTMS + K group) included 35 patients with confirmed C4-Th12 spinal cord injury. The group that received only kinesiotherapy consisted of 55 patients with C4-Th12 spinal cord injury (K group). The control group of healthy volunteers (N = 45, 15 women and 30 men) was examined to obtain the reference values of control neurophysiological recordings. The age of the healthy group was 37.9 ± 4.0 (range from 24 to 51), their height was 161–180 cm with a mean of 168.1 ± 3.6 cm, and their weight was 51–81 with a mean of 60.7 ± 5.3. The patients and healthy volunteers did not differ significantly in terms of age, height, and weight (*p* = 0.7). We did not find differences in age, height, and weight between groups of patients with *p* < 0.05.

The general practitioner, neurosurgeon, neurologist, and neurophysiologist evaluated the health status of both controls and patients. 

The majority of our patients suffered an incomplete spinal cord injury that occurred in a traffic accident, which is confirmed by data from the literature that clearly indicate traffic crashes as the most common cause [18]. All patients were treated neurosurgically (spinal stabilisation), and their general health was stabilised before they started rehabilitation. Clinical studies were conducted according to the American Spinal Injury Association (ASIA) impairment scale, and the tests revealed AIS C and D in patients in all five studied groups prior to rehabilitation in almost equal proportions (Table 1). Both grades stand for the preservation of sensation to a greater extent and incomplete motor impairment, but the difference is that at least half of the key muscles below the level of injury are not strong enough to move against gravity (a muscle grade less than 3) in the case of the ASIA C, and at least half of the key muscles below the level of injury are strong enough to move against gravity (a muscle grade is 3 or more) in the case of ASIA D.

The preservation of the spinal cord structure in 1/3 to 1/4 verified by MRI was the main inclusion criterion. It was also important to us that patients enrolled in the study were no less than 4 months but no more than a year and a half after injury (Table 1). The average observation time before–after therapy in all groups of patients treated with the different modalities which were similar to the duration of the applied therapy lasted about 8.6 months in rTMS + K, K, rTMS, and Electro and about 9.9 months in Electro + K. The main exclusion criteria from the study were contraindications to the use of stimulation, i.e., pregnancy; severe head injury; episodes of epilepsy; severe disorders of the cardiovascular system; electronic implants and devices such as a pacemaker, an insulin pump, a baclofen pump, or a cochlear implant; stroke; episodes of plexopathies during treatment; inflammatory diseases; and myelopathies before or diagnosed after the incident. Patients who were ineligible for stimulation treatments but agreed to participate in the study were included in the kinesiotherapy-only treatment group. All the patients were informed and understood the potential for no benefit and the risk of all the procedures. The study was conducted according to the guidelines of the Declaration of Helsinki and was approved by the Bioethics Committee of the Medical University (decision no. 942/2021).

### 2.2. Kinesiotherapy

There is no universal, recommended approach to the rehabilitation of patients with spinal cord injury in the literature [19]. This is mainly due to the different levels of damage, the severity of neurological dysfunction, and the different clinical condition of patients both before and after the injury. Intensive and regular exercise supervised by physiotherapists is the cornerstone of patient rehabilitation, but a kinesiotherapy program must be tailored individually. The stimulation methods (rTMS and electrical stimulation) described in this paper that can support neuroplasticity can only be an adjunct to kinesiotherapy but should never replace it. 

The spinal cord injury patients from our study underwent a rehabilitation program created and developed over the years in the Neurorehabilitation Center for Treatment of Spinal Cord Injuries AKSON, Wroclaw, Poland. It is one of the few such places in the country dedicated specifically to patients who have suffered spinal cord injuries, being staffed by physiotherapists who constantly consult with neurosurgeons, neurologists, and neurophysiologists. There, exercises are individually tailored to patients by qualified physiotherapists, but there are a few universal rules—patients exercise for three months for several hours a day (4–6 h per day, five days a week), and these are very intense workouts in which a patient is an active participant. The programme includes postural stabilisation exercises, exercises without and with loadings (Figure 1D), stretching, and learning daily activities. The effectiveness of rehabilitation is not only assessed clinically but also by neurophysiological tests, for which are able to show even subclinical changes in voluntary motor function (e.g., clinically, the movement in the patient’s extremity is not detected, but there is observed an increase in amplitudes in sEMG recording, which indicates better improvement in recruitment of the muscle motor units). When asked to contract a muscle, patients can see a change in the sEMG recording on the recorder’s screen. Such biofeedback helps patients to see their progress and motivate them to work further. 

Most of the rehabilitation specialists agree that patients with iSCI should start physical therapy as early as post-injury day one, and it is recommended to provide at least a once-daily session with the target goal being >20 min of maximum-tolerated activity. Thus, we are well aware that the very intensive rehabilitation program that our patients undergo [14,15,20] is much more than the standard and average kinesiotherapy algorithm [20]. 

### 2.3. Transcranial Magnetic Stimulation (TMS)

Transcranial magnetic stimulation (TMS) is a method of non-invasive stimulation through the intact scalp. The magnetic field evokes an electrical response in the cortical neurons [10,15]. This method is very commonly used in various psychiatric disorders. TMS is also used in diagnostics to produce motor-evoked potentials (MEP) to test the functional integrity of the corticospinal tract [15]. By stimulating the motor centre, it is verified if the impulse is conducted from the brain through the spinal cord to the peripheral nerves and effectors. MEP parameters such as amplitudes, latencies, and response morphology are very useful diagnostic tools that help to reveal abnormalities in efferent transmission [21]. However, to our knowledge, we are one of the very few research teams in Poland (represented by clinicians from Poznań and Wrocław) that uses and analyses rTMS (repetitive transcranial magnetic stimulation) in the therapeutic (not only diagnostic) process of patients with spinal cord injuries. TMS is restricted in those people having magnetic implants, events of epilepsy, or recent adverse neurological or cardiac events [22]. Using such exclusion criteria, it is in vain to find reports in the literature of serious side effects of this therapy, which is considered safe and non-invasive.

Non-invasive, repetitive transcranial magnetic stimulation increases descending activity so isolated neuronal circuits below the injury can receive supraspinal input. It may induce cortical and spinal cord plasticity and restore connectivity between cells [23]. The remaining connections are strengthened by rTMS, which helps the nervous system to use these surviving connections and rewire itself. Magnetic stimulation, unlike electrical stimulation, is painless, provided the appropriate parameters are used. Electrical stimulation requires large electric currents, which may cause very painful and unpleasant contractions of scalp muscles and sensory reception activation [21]. This is why magnetic stimulation seems to be more suitable for clinical use. This stimulation of the motor cortex and elicitation of a response in the muscle in this way is painless; non-invasive; and, in our experience, well tolerated by iSCI patients. It also does not require a surgical procedure like direct epidural spinal cord stimulation. It is, therefore, simpler and less expensive to apply [5,19].

In the literature, we can find various papers evaluating the use of TMS on small groups of patients and with different algorithms. On the basis of the available literature, we created our own stimulation algorithm, which differs primarily in the applied intensity of stimulation. 

The rTMS study tested the effect of treatment using transcranial magnetic stimulation in patients after iSCI. Therapy lasted from 2 to 14 months, with an average of 8 months, and consisted of 3–5 sessions per month with a maximum of 15 sessions in the whole treatment period. The differences in length and frequency were mostly due to the social reasons of the patients. The devices we use for both rTMS and diagnostic MEPs are the MagPro R30 and MagPro X100 magnetic stimulator with MagOption (Medtronic A/S, Skøvlunde, Denmark) (Figure 1B). The motor cortex area with greater representation for the lower limbs has been stimulated with a 15–25 Hz frequency and pulse strength that has been 70–80% of a patient’s resting motor threshold (RMT), usually not more than 45% of the maximal device’s output. Patients receive 800 pulses per hemisphere. Trains last 2 s and consist of 40 pulses. Trains are separated by 28 s intervals. These are the fixed parameters of our algorithm. However, patients with poorer neurophysiological test results (e.g., lower amplitudes in sEMG recordings) have higher frequency values applied. An individual RMT is determined by the diagnostic MEPs performed prior to the stimulation. Every patient may have a different RMT because it is the minimal stimulation intensity needed to evoke a reliable motor response seen as a twitch in a tested muscle or, if the muscle contraction is unnoticeable, >50 μV peak-to-peak amplitude above the background electromyographic activity. On the basis of the RMT determined in this way, the stimulation strength is then set, which in the case of our algorithm, does not exceed 70–80% of the individual RMT. For stimulation, we used a circular coil (C-100, 12 cm in diameter) (Figure 1B(b)). We applied it over the scalp at the expected location of the motor cortex, previously confirmed by performing MEP (Figure 1 B(a)). It is the anatomical origin of the neurons and fibres of origin of the descending corticospinal pathway.

### 2.4. Peripheral Electrotherapy

The electrical stimulation applied to the nerves is more commonly used in the rehabilitation of patients. It is more readily available, less expensive, and more common in clinical use to aid neuroplasticity processes [10], but, for some reason, is still rarely used in patients after spinal cord injuries. There is a poor understanding of the mechanisms underlying the improvement of neurological function, but it is theorised that peripheral stimulation may prevent increased levels of neurotrophic factors, stimulate neuronal outgrowth, and increase the excitability of neuronal networks below the lesion; thus, muscle atrophy can be prevented [10,24,25]. This seems to be a very important safeguard against secondary pathological changes in SCI patients that appear in motor fibres of lower extremities despite no direct damage to the lumbosacral region of the spine [16]; however, there are a lack of data on this issue in the literature because most of the studies are conducted on animals, rarely on humans, and using neurophysiological tests as the feedback [26,27,28,29,30].

We performed an individually adjusted, home-based electrostimulation dedicated to the peroneal and tibialis nerves. The device is a personal, mobile, four-channel stimulator. Patients were given a portable device to take home, on which the stimulation algorithm was programmed (NeuroTrac^®^ Sports XL, UK). They were also given precise instructions on how to use the device and how to apply the two pairs of self-adhesive stimulating electrodes (Axelgaard Ultrastim Wire Neurostimulation Electrodes with MultiStick Gel, 5 × 5 cm, UK), which were placed bilaterally in the area of the popliteal fossa on the anatomical passage of tibial and peroneal nerves, with the anode placed more proximal to the cathode (Figure 1C). Bipolar, rectangular electric pulses were applied in series. Stimulus strength was the only parameter that was able to be adjusted by patients. Patients were instructed that they could increase the strength of the stimulus on their own until they could see the toe move but not yet feel pain. All the rest parameters, such as frequency (20–70 Hz), single stimulus duration (11–22 ms), train duration (2–3 s), the interval between trains (2–3 s), and session duration (15–20 min) were programmed by the neurophysiologists and tailored to the patient on the basis of their electromyography and electroneurography recordings conducted prior to the treatment. Patients were unable to change these parameters themselves. Patients with more severe neurogenic changes had a longer single stimulus duration and lower frequency set. The stimulator made it possible to read the actual stimulation time used by the patient. This was an additional incentive for patients to follow the recommendations because they knew that whether they followed the plan or not was verified. Patients followed the stimulation regime, as evidenced by the very similar actual stimulation times read from the stimulators to those we expected from the patients.

### 2.5. Neurophysiological Feedback

We compared the effectiveness of both stimulation methods using bilateral surface electromyography (sEMG) to evaluate motor unit recruitment in the lower extremities (Figure 1A(a–c)). We took the tibialis anterior (TA) as the index muscle. sEMG was performed before and after the applied treatment using the KeyPoint Diagnostic System (Medtronic A/S, Skøvlunde, Denmark), and all patients were lying in the supine position. We applied standard, disposable Ag/AgCl surface recording electrodes (5 mm^2^ of an active surface) with an active electrode placed on the muscle belly, a reference electrode placed on the distal tendon of the same muscle, and a ground electrode placed on the distal part of the leg—according to the Guidelines of the International Federation of Clinical Neurophysiology—European Chapter [15,16]. We set the upper 10 kHz and the lower 20 Hz filters in the recorder. The sEMG examination consisted of two stages. First, we asked the patient to lie down comfortably and try to relax the muscles of the lower limbs as much as possible (Figure 1A(b)). Then, the patients were instructed to contract the muscles under examination, and, on the command of the examiner, the patient made the strongest possible contraction of the muscles in the lower limbs and maintained it for 5 s. They performed 3 attempts each time, separated by 1 min rest. The examiner selected the best attempt for analysis—the one with the highest mean amplitude (in μV)—peak-to-peak with reference to the isoelectric line. The output measures were the parameters of the amplitude measured in μV and the frequency of muscle motor unit action potential recruitment measured in Hz. Frequency index (3–0) was scored on the basis of the calculations of the motor unit action potential recruitment during maximal contraction in sEMG recording: 3 = 95–70 Hz—normal; 2 = 65–40 Hz—moderate abnormality; 1 = 35–10 Hz—severe abnormality; 0 = no contraction. sEMG recordings were performed at the base time of 80 ms/D and the amplification of 20–1000 μV/D (Figure 1E–H).

### 2.6. Statistical Analysis

Data were analysed with Statistica, version 13.1 (StatSoft, Kraków, Poland). Descriptive statistics included mean values, standard deviations (SD), and minimum (min) and maximum (max) values for measurable variables. The normality distribution and homogeneity of variances were conducted with Shapiro–Wilk tests and with Levene’s tests in some cases. In patients of five groups with incomplete spinal cord injuries at the cervical and thoracic levels, the mean values of parameters from sEMG tests were compared before and after therapy using Student’s *t*-test, Welch’s test, the Mann–Whitney test, or Wilcoxon’s test. It was assumed that a comparison of values at *p* ≤ 0.05 determined significant statistical differences. Changes in the outcome measures from sEMG recordings (parameters of amplitudes and frequency indexes) before and after treatment were also expressed in percentages. The preliminary statistical analysis determined the required sample size using the primary outcome variables from sEMG recordings from anterior tibialis muscles before and after treatment with a power of 80% and a significance level of 0.05 (two-tailed). The mean and standard deviation (SD) were calculated using the data from the first ten subjects. The sample size software estimated that at least 25 subjects from each of the studied groups were needed.

## 3. Results

None of our patients reported side effects from stimulation, whether rTMS or peripheral electrotherapy.

Subjects belonging to the five groups of iSCI patients treated with different methods did not differ greatly in terms of demographic and anthropometric characteristics, as well as the severity and extent of spinal cord injuries (Table 1). Although the number of patients in each group was different, all subjects represented similar injury levels and ASIA scores. The differences in weight, height, time from injury, and observation time were either not significant or at the limit of the statistical significance (*p* = 0.05). It is not likely that these variables and factors influenced the differences found in the comparison of neurophysiological results in the studied groups of patients (Table 2).

Table 2 shows a summary of the sEMG parameters recorded from the anterior tibialis muscle in iSCI patients belonging to five therapeutic groups before and after the applied treatment. All of them differed from those recorded in a control group of healthy volunteers (Table 3), both with references to the recorded values of amplitudes and the frequencies of firings of motor units’ action potentials (Figure 1E–H). After the therapies were applied, a significant improvement of the mean amplitudes in sEMG recordings from anterior tibialis muscles were found in patients from the rTMS + K group (at *p* = 0.008) by 32.1% and Electro + K group (at *p* = 0.009) by 44.4% in comparison to the values recorded before therapy. A significant increase at *p* = 0.03 of the sEMG amplitude parameter after therapies was observed in patients belonging to the Electro (27.5%) and rTMS (26.4%) groups. A significant improvement (at *p* = 0.04) in the frequency index of sEMG recordings after the therapy was observed in patients from the Electro + K group by a 22.2% difference, although by 16.6% was also ascertained in patients treated with electrotherapy only, and by 11.7% in rTMS + K and rTMS groups. A slight difference in the improvement of the amplitude parameter by 14.8%, without a significant improvement in the frequency index, was found in the group of patients treated only with kinesiotherapy (K).

In all the groups of patients treated with five types of therapies, there were no observed improvements in the parameters of the amplitude and frequency of sEMG recording from the anterior tibialis muscle during the attempt of maximal contraction comparable to those recorded in healthy volunteers from the control group (Table 3).

## 4. Discussion

This study compared the effectiveness of three different treatment options, namely, combinations of rTMS, peripheral electrotherapy, and kinesiotherapy, evaluated with clinical neurophysiology measurements in patients with iSCI. Our data show that better improvement of the average values of sEMG amplitudes more than firing frequencies of motor units action potentials recorded from the anterior tibialis muscle and better general neurophysiological improvement were found in iSCI patients treated with peripheral electrotherapy combined with kinesiotherapy and rTMS combined with kinesiotherapy. Patients treated with kinesiotherapy alone achieved worse results. 

Table 4 and Table 5, with the data on the algorithms of rTMS and peripheral electrotherapy described in the literature, show how the parameters and methods used to evaluate therapy varied in patients after iSCI.

In the literature, there are many examples of rTMS applications in the treatment of psychiatric disorders [33], as well as the use of TMS as a diagnostic tool that, by means of MEP, reflects the integrity of the corticospinal tract and the impulse conduction from the motor cortex to the muscles [34,35]. However, there are still very few scientific reports using rTMS as an adjunct to treatment in patients with spinal cord injury [11,12,13,14,31,32]. The effect of rTMS stimulation on the cortical structure function depends mainly on the frequency of pulses during stimulation. The high frequency used in rTMS (>10 Hz) is known as excitatory. It increases activity in the neurons of the corticospinal pathway and also reduces corticospinal inhibition. Therefore, we use frequency values above 20 Hz in therapy, as opposed to low frequencies, which would have the effect of reducing cortical excitability [15]. Compared to the therapy algorithms used in the literature (see Table 4), we used a much lower stimulation strength—70–80% of the individually selected RMT, which is no more than 0.8–1 Tesla (usually 38–45% of the maximum device’s output and not close to 90–100% as in other studies). In our opinion, such a low stimulus strength is still effective; it stimulates structures up to 3–5 cm deep into the cortex, but at the same time, it is less painful for the patient, which has an impact on engagement in therapy. Stimulus strength of more than 100% RMT (80% and more of the maximum device’s output) when performing diagnostic MEPs is painful for patients. Not to mention such high stimulus values applied in series and repeated hundreds of times during rTMS. Such painful therapy does not have a positive effect on patient compliance. In addition to other studies, we used a significantly longer observation period.

Perez et al. [35] presented their study on healthy humans, which indicates the presence of short-term plasticity within spinal inhibitory circuits under the influence of electrical nerve stimulation. They hypothesised that the pattern of electrical input might be a crucial factor for inducing changes in the spinal circuit and that such stimulation may improve walking ability in patients with spinal cord injury [35]. However, in the case of currently used peripheral electrotherapy, it is very common to use the same stimulation parameters without neurophysiological-based individual adjustment [29]. Moreover, these algorithms differ greatly in their values and in the way they are assessed. Single stimulus duration may vary from 0.1 to 500 ms, frequency 1 to 100 Hz, and intensity is set up even to 250 mA [36]. Our algorithm has parameter ranges that are individually tailored to the patient on the basis of their neurophysiological test results (see Table 5). As with rTMS, here, too, we performed a much longer observation period. 

A very interesting summary of the literature on the use of TMS as a therapeutic method was used by Brihmat et al. [37]. He and his team provide a very detailed analysis of various stimulation algorithms and their evaluation methods. The conclusions are similar to ours—there are still many gaps in the rTMS algorithms presentation and, in general, in SCI research. This summary also confirms that TMS is a minimally invasive, safe, and easy-to-use method that may provide various positive clinical and neurophysiological results and should be considered in the rehabilitation of the SCI population [37].

The activity of the anterior tibial muscle during the maximal contraction used as the marker of motor improvement following the application of different rehabilitation procedures in iSCI patients was chosen on the basis of the suggestions of other researchers [38] as well as previous experiences [9,15,16]. During electrotherapy procedures, we stimulated peroneal nerves innervating tibial muscles, for which sEMG were recorded [16]. We concluded in the previous study that the activity of thigh muscles (e.g., rectus femoris muscles) was usually better in iSCI patients because they are innervated by femoral nerves whose trunks are larger and include a greater spectrum of axons than peroneal nerves [9]. The secondary, extensive degenerating processes in motor fibres and muscle atrophy are clearly detected in distal than proximal parts of nerves of the lower extremities [16,25]. Therefore, in cases of patients after iSCI one year after the incident, it is recommended to choose sEMG diagnostic recordings from the tibial muscle instead of the foot extensors, although they are both innervated from peroneal nerves [19,38].

The increased amplitude (in μV), more than the frequency index (in Hz) parameters, in the sEMG recording after the applied treatment (see Table 1) indicates better recruitment of motor units of the examined muscles after therapy. Such neurophysiological improvement is not necessarily always simultaneous with clinical improvement, but it certainly indicates positive changes in the muscles of the lower extremities in the patients treated with neuromodulation methods. Such changes may also correlate with the improvement of the transmission of neural impulses. The question arises, is any of these stimulation methods better? Our data show that better average sEMG amplitudes for the anterior tibialis muscle were recorded by patients treated with peripheral electrotherapy and kinesiotherapy. However, it should be noted that each of these stimulation methods has a different mechanism for improving patients’ neurophysiological status and neuroplasticity, so it may be beneficial to use both methods, if possible. However, this hypothesis requires further research. Nevertheless, our results clearly indicate that the use of even one of the methods of stimulation, regardless of the supraspinal and spinal or peripheral mechanism, provides better results than kinesiotherapy alone. Due to the length of the therapy (several months) and the strenuousness of the repeated stimulation and neurophysiology tests, we took great care to ensure that these procedures were as painless as possible for patients and as little traumatising as possible, having in mind patients’ comfort and well-being. Hence, the algorithms with a much lower electrical and magnetic stimulation intensity than those used by other researchers (see Table 4 and Table 5) and the use of surface electromyography (sEMG) instead of invasive needle electromyography for evaluation of the treatment results were considered. For diagnostic purposes, sEMG is contemporarily considered a functional test precise and specific enough to evaluate the entire efferent transmission from the supraspinal level to the muscles in iSCI patients treated with the different classical conservative methods as well as the newest based on the neuromodulation [38,39]. Nevertheless, a key difference in the use of the rTMS and electrical stimulation of nerves is their availability. Patients qualified for rTMS had to attend the treatment facility regularly for stimulation application. This requires more time and financial investment in terms of travel. Patients qualified for electrotherapy applied stimulation independently at home. This, too, may have affected the patient’s involvement in the therapy process.

In looking for ways to improve the functional status of people after an incomplete spinal cord injury, it is important to remember that the dysfunction is not only due to primary injury, local synaptic and neuronal loss, inflammation, and the later formation of scar tissue, which are most often treated with pharmacology and surgery. The main problem for these people becomes a change in the flow of information in ascending and descending pathways. Over time, the neuronal projections become weaker, and this may lead to secondary degenerative changes in the subcortical and cortical structures and brain reorganisation [3,5,40]. Therefore, it seems reasonable to look for stimulation methods that enhance activity-dependent neuroplasticity and residual signals and alter neuronal activity [10,41]. Such a method may support neurorehabilitation by fulfilling the premise of Hebb’s theory—neurons that fire together wire together [42,43]. Coordinated activity of a presynaptic terminal and a postsynaptic cell would lead to the strengthening of that synaptic connection between them, and conversely, uncoordinated activity, which is observed in iSCI patients, between synaptic partners would weaken their synaptic connections. It speaks powerfully about the consequences of coordinated activity for not only synaptic strength but also the long-term consequences that require the building up of new synaptic connections. Currently, many different methods of stimulation are being tested, and hopes are being held that a proper stimulation algorithm can modulate excitability; thus, we may be able to establish new synaptic connections, improve neural transmission, and inhibit and reverse the degenerative process in axons. Stimulation methods such as transcranial magnetic stimulation and peripheral electrotherapy should be included in the therapy in the first year after the injury because the first six months to a year appears to be a crucial time period for spinal cord regeneration; structural and functional reorganisation of the spine; and the brain and, thus, functional recovery [9,11,44,45,46]. However, we still need more evidence because the mechanisms underlying stimulus-induced recovery are not well understood.

Incomplete spinal cord injury is a pathological complex syndrome in which the treatment can be disturbed with many factors such as chronic non-bacterial osteomyelitis [47], and hence many modern surgical [48] and non-surgical [49] therapeutic methods are incorporated. The main problem in the management and treatment of patients with spinal cord injuries is the use of different algorithms and methods of evaluating these patients and the lack of consistency between observed effects [9]. The stimulation algorithms presented in Table 4 and Table 5 are difficult to compare because the authors use different scales, such as the ASIA scale, the visual analogue scale of pain (VAS), the Ashworth scale, and different clinical assessment tools that are subjective in nature, thus making it especially challenging to report the results of applied therapy, interpret them in a consistent manner, and compare them across studies. In this work, we would like to emphasise the important role of neurophysiological tests such as electromyography, eletroneurography, and motor-evoked potential examinations. These methods allow clinicians to assess the progress of patients’ physical recovery objectively; therefore, in our opinion, they should be an obligatory part of the assessment of patients after spinal cord injury. Such methods of assessing the effects of treatment will make it possible to compare further results, which will contribute to providing more evidence of the effectiveness of applied treatment algorithms. This will make discussions between groups of researchers and the comparison of research results much easier. Moreover, such evaluation will also make it possible to assess subclinical changes that are impossible to capture by the scales commonly used in the neurological evaluation of patients and standard physical examination. Capturing changes (even subclinical), such as an increase in amplitudes in sEMG or in MEP responses, is evidence of improvement and can be a signal that the treatment algorithm used is heading in the right direction, which, especially for patients, is a follow-up and great motivation for further work and rehabilitation. This is of utmost importance in patients for whom spinal cord injury is a life-altering trauma; thus, they look for any improvement in their health status. The use of this neurophysiological test not only allows for the comparison of the effectiveness of different treatment options (such as the comparison presented in this study) but also allows for the modification and personalisation of the process, bringing us closer to creating an evidence-based rehabilitation algorithm.

The challenge and the main limitation of all research related to spinal cord injury is the highly heterogeneous pathophysiology and different dynamic of the spontaneous regeneration process [5]. Most cases involve incomplete injury, but even in this group, the severity depends on the level of damage and which parts of the spinal cord are affected. Such a diversity of cases and pre- and post-injury health status makes it very difficult to compare patients. One must always take this into account in studies and try, as in our work [15,16], to include patients with damage at a similar level and with similar neurophysiological status. Unlike other authors [12,13,14,30,33], for our study, we did not use sham stimulation. Moreover, as it appears in the data presented in Table 5, the protocols for electrical stimulation parameters are different in different studies regarding muscle and nerve electrotherapy. Therefore, they cannot be sufficiently compared and justified. We believe that the small number of spinal cord injury patients who are eligible for stimulation and are willing to participate in several months of therapy deserve to be given every possible chance for treatment. Such people, seeking every possible chance of therapy and functional recovery, would find it difficult to face a possible placebo approach. For this reason, we compared patients treated only with kinesiotherapy who were unwilling or ineligible for additional therapy with patients receiving additional stimulation treatment. We are aware of the scientific downside of such an approach, but ethical considerations prevail here to ensure that our patients receive the best possible treatment algorithm, especially in a tragic event such as a spinal cord injury, which is life-changing not only for the patient but also for those around them.

We hope that the evidence gathered here on the effective application of transcranial magnetic stimulation or peripheral electrostimulation as methods facilitating functional recovery in patients with spinal cord injuries will encourage clinicians to use these methods, evaluate their effectiveness with neurophysiological tests, and present further results.

## 5. Summary of Recommendation

Rehabilitation should be started as soon as possible—it should be intensive and individually tailored to a patient’s needs. It is a foundational and indispensable element of every treatment program for any patient after iSCI, regardless of the level of damage. Kinesiotherapy effects alone are not satisfactory enough. The treatment should include repetitive transcranial magnetic stimulation or peripheral electrotherapy or both in the first year after injury—the first six months to a year appears to be a crucial time period for spinal cord regeneration and functional recovery. The possible algorithms are proposed in Table 4 and Table 5. The effects of therapy should be evaluated not only with the clinical methods but using objective clinical neurophysiology tests such as electromyography, electroneurography, and motor-evoked potentials examinations. Observation of even the subclinical changes expressed in improvements in clinical neurophysiology tests can be an additional motivation for patients to continue the rehabilitation programme.

## 6. Conclusions

The concept of neuromodulation, aiming to improve the activity in spinal centres of iSCI patients, assumes a combination of afferent pulsation (provided by electrotherapy or to some degree by different procedures of kinesiotherapy) and efferent (provided by rTMS) to obtain the facilitated response in general exemplified in improving motor function according to Hebb’s theory [42]. Since the results of treatment in two more groups of patients only with rTMS and only with nerve electrotherapy were presented, the effectiveness of these technologies for the treatment of spinal cord injury consequences can be better understood. Transcranial magnetic stimulation and peripheral electrostimulation appear to be methods that support and facilitate the recovery process in iSCI patients. Both simulation methods should be complementary to standard kinesiotherapy since they improve patients’ neurophysiological status and provide better sEMG results than kinesiotherapy alone. In addition, the entire neurorehabilitation process should be evaluated by objective methods of clinical neurophysiology, such as electromyography. Such a method of assessing the effects of treatment will make it possible to compare further results, which will contribute to providing more evidence of the effectiveness of applied treatment algorithms. Moreover, such evaluation will also make it possible to assess subclinical changes that are impossible to capture by the scales commonly used in the neurological evaluation of patients and standard physical examination. All of these allow us to modify and personalise the management of spinal cord injury patients and may bring us all closer to the main task ahead, which is establishing an evidence-based neurorehabilitation program for subjects after iSCI.

## Figures and Tables

**Figure 1 biomedicines-11-01035-f001:**
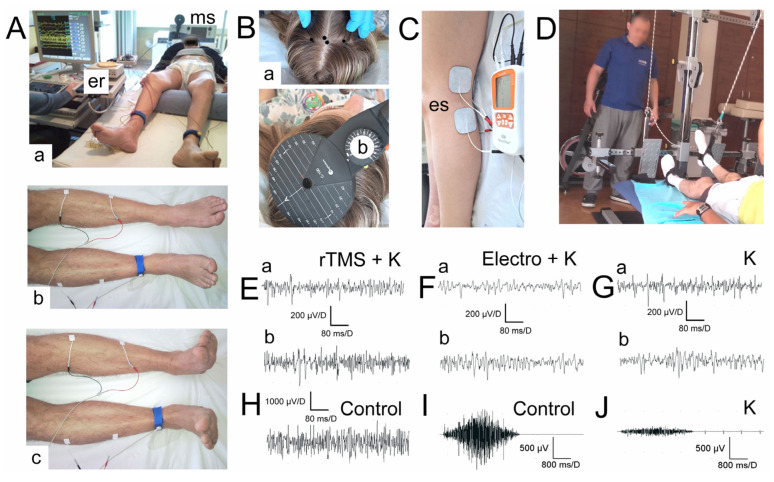
Photographs illustrating methodological principles of sEMG recordings (**A**): (a) electromyography recorder “er” and the magnetic stimulation device “ms”; (b) bilateral location of sEMG bipolar surface electrodes during recordings from tibialis anterior muscles at rest; (c) sEMG recording during the attempt of 5 s lasting muscles maximal contraction and applied therapies in five groups of iSCI patients. (**B**) rTMS or rTMS + K—groups of patients treated with rTMS and additionally with kinesiotherapy: (a) marking of the “hot spots” for repetitive transcranial magnetic stimulations; (b) location of coil during releasing the trains of stimuli from the magnetic stimulation device “ms”. (**C**) Electro or Electro + K—groups of patients treated with electrotherapy and additionally with kinesiotherapy (“es”)—bipolar electrode placement and the device for functional electrical stimulation during nerves electrotherapy. (**D**) K—patients treated with kinesiotherapy only (system for lower extremities strengthening exercises with loadings is shown). (**E**–**G**) Examples of sEMG recordings from anterior tibialis muscles during the attempt of maximal contraction in patients belonging to three studied groups before (a) and after (b) therapies. sEMG recording from one of the healthy volunteers is shown in (**H**) for comparison. (**I**,**J**) Examples of sEMG recordings presenting 5 s lasting maximal contraction with a slower time base in one of the subjects from the control group and a patient from the (**K**) group, respectively. Calibration bars for different amplifications and time bases are presented.

**Table 1 biomedicines-11-01035-t001:** Demographic, anthropometric, and traumas characteristics as well as the average duration of the applied therapies (before–after observation time) in the iSCI patients treated with different modalities. Minimum and maximum values, mean values, and standard deviations are presented.

Variable Patients Group	Age (Years)	Height (cm)	Weight (kg)	Averaged Time from Injury (Months)	Averaged Observation Time (Before–After) (Months)	ASIA AIS Score	Injury Level
**rTMS + K** 12♀, 23♂ N = 35	23–51 38.1 ± 3.5	157–178 170.1 ± 2.9	50–86 60.7 ± 4.1	6.7 ± 1.4	8.9	C = 26 D = 9	C4–C8 = 18 Th1–Th12 = 17
**Electro + K** 23♀, 42♂ N = 65	25–56 37.2 ± 4.1	161–184 172.1 ± 3.6	50–85 62.1 ± 4.3	7.4 ± 1.9	9.9	C = 54 D = 11	C4–C8 = 29 Th1–Th12 = 36
**K** 18♀, 37♂ N = 55	24–53 37.4 ± 3.6	159–183 173.2 ± 3.6	49–84 63.5 ± 3.7	7.1 ± 1.8	8.7	C = 46 D = 9	C4–C8 = 27 Th1–Th12 = 28
**rTMS** 12♀, 22♂ N = 34	25–58 38.1 ± 3.3	163–180 174.2 ± 3.2	51–84 64.6 ± 3.7	6.9 ± 1.6	8.4	C = 24 D = 10	C4–C8 = 15 Th1–Th12 = 19
**Electro** 19♀, 34♂ N = 53	24–60 38.1 ± 4.0	158–179 175.4 ± 3.8	50—80 62.8 ± 3.6	6.7± 1.2	8.7	C = 45 D = 8	C4–C8 = 24 Th1–Th12 = 29
*p*-value (difference) rTMS + K vs. Electro + K Electro + K vs. K rTMS + K vs. K rTMS + K vs. rTMS rTMS vs. Electro rTMS vs. K Electro vs. K	NS NS NS NS NS NS NS	NS NS 0.05 0.05 NS NS 0.05	0.05 NS 0.05 0.05 0.05 NS NS	0.05 NS 0.05 NS NS NS NS	NS 0.05 NS NS NS NS NS	NA NA NA NA NA NA NA	NA NA NA NA NA NA NA

Abbreviations: rTMS + K—a group of patients treated with rTMS and kinesiotherapy, Electro + K—a group of patients treated with electrotherapy and kinesiotherapy, K—patients treated with kinesiotherapy only, rTMS—patients treated with rTMS only, Electro—patients treated with electrotherapy mainly, NS—non-significant, NA—non-applicable.

**Table 2 biomedicines-11-01035-t002:** A summary of the sEMG parameters measured from the anterior tibialis muscles (cumulative data for the right and left sides) in the iSCI patients belonging to five therapeutic groups before and after the applied treatment. *p*—significant differences are marked in bold. %—the percentage of change.

Groups of Patients	rTMS + K		Electro + K		K		rTMS		Electro	
Parameter	Before	After	Before	After	Before	After	Before	After	Before	After
TA sEMG Amplitude (μV) Frequency index	50–2100 232.7 ± 94.5 1–3 1.5 ± 0.6	50–2300 343.0 ± 79.2 1–3 1.7 ± 0.5	50–2000 191.4 ± 94.7 1–3 1.4 ± 0.5	50–2400 344.2 ± 88.3 1–3 1.8 ± 0.5	50–1900 221.0 ± 87.1 1–3 1.4 ± 0.5	50–2000 259.3 ± 93.8 1–3 1.5 ± 0.6	50–2000 229.3 ± 83.1 1–3 1.5 ± 0.4	50–2300 311.9 ± 81.6 1–3 1.7 ± 0.6	50–2000 210.5 ± 85.6 1–3 1.5 ± 0.5	50–2300 290.6 ± 82.1 1–3 1.8 ± 0.5
Neurophysiological improvement (before vs. after) Amplitude (μV) Frequency index	**0.008** **0.05**	32.1% 11.7%	**0.009** **0.04**	44.4% 22.2%	**0.05** NS	14.8% 6.6%	**0.03** **0.05**	26.4% 11.7%	**0.03** **0.05**	27.5% 16.6%

Abbreviations: rTMS + K—a group of patients treated with rTMS and kinesiotherapy, Electro + K—a group of patients treated with electrotherapy and kinesiotherapy, K—patients treated with kinesiotherapy only, rTMS—patients treated with rTMS only, Electro—patients treated with electrotherapy mainly, Frequency index (3–0)—frequency of motor unit action potentials recruitment during maximal contraction sEMG recording: 3 = 95–70 Hz—normal; 2 = 65–40 Hz—moderate abnormality; 1 = 35–10 Hz—severe abnormality; 0 = no contraction; NS—non-significant.

**Table 3 biomedicines-11-01035-t003:** Reference values of surface electromyography (sEMG) parameters that were recorded during an attempt of maximal contraction in a group of healthy volunteers (N = 45).

Recording	Parameter	Healthy Volunteers (Control)	
TA sEMG	Amplitude (µV)	600–2450	785.3 ± 100.2
	Frequency index	3–3	3.0 ± 0.5

Abbreviations: TA—anterior tibialis muscle; sEMG—surface electromyography. Frequency index (3–0)—frequency of motor unit action potentials recruitment during maximal contraction sEMG recording: 3 = 95–70 Hz—normal; 2 = 65–40 Hz—moderate abnormality; 1 = 35–10 Hz—severe abnormality; 0 = no contraction.

**Table 4 biomedicines-11-01035-t004:** Data on parameters of transcranial magnetic stimulation in patients with spinal cord injury were gathered from available literature and from observations where effects confirmed its efficiency.

Reference	No of Patients Who Received Stimulation	Stimulation Algorithm	Interval between Trains of Stimuli	Intensity of Stimulation, Maximal % Stimulus Output	Number of Sessions Per Day, Week, Month	Evaluation Method of Effectiveness
**Belci** **et al.,** **2004 [11]**	4	10 Hz 720 pulses in total	100 ms	90%	5 sessions of 1 h, one per day	ASIA scale 5 days of sham rTMS delivered over the occipital cortex
**Kumru** **et al.,** **2010 [12]**	15	20 Hz 40 pulses in a train 1600 pulses in total	28 s	90%	15 sessions, 5 per week, 3 weeks of observation in total	Clinical evaluation of spasticity, EMG Sham stimulation with a coil disconnected from the main stimulator unit
**Kuppuswamy** **et al.,** **2011 [13]**	15	5 Hz 900 pulses in total	8 s	80%	5 days, 1 per day (sham or active, randomly)	ASIA scale, upper limb functional tests, MEP, autonomic measures; sham stimulation with only 5% of real stimulator output
**Benito** **et al.,** **2012 [14]**	7	20 Hz 40 pulses in a train, 1800 pulses in total, 20 trains	28 s	90%	15 days, 1 per day (sham or active)	Functional gait assessment and clinical spasticity assessment Double-blind, sham-controlled trial; sham stimulation with a coil disconnected from the main stimulator unit
**Nardone** **et al.,** **2014 [31]**	9	20 Hz 1600 pulses in total	28 s	100%	4 days, 1 session per day	Clinical evaluation of spasticity, EMG Sham stimulation with a coil disconnected from the main stimulator unit
**De Araujo** **et al.,** **2017 [32]**	10	5 Hz 50 pulses in a train 12 trains	10 s	100%	10 sessions over 2 weeks	ASIA scale, Fugl–Meyer scale, Modified Ashworth Scale, socio-demographic questionnaire, Mini-Mental State Examination, EMG, EEG 5 sessions active and 5 blind for each person, with a two-week washout period
**Wincek** **et al.,** **2021 [32]**	26	15–25 Hz 40 pulses in a train, 20 trains 1600 pulses in total	28 s	38–40%	3–5 sessions per month for 5 months	sEMG, MEP; results were compared to patients treated with kinesiotherapy alone

**Table 5 biomedicines-11-01035-t005:** Data on parameters of peripheral electrostimulation in patients with spinal cord injury gathered from the available literature and from observations where effects confirmed its efficiency.

Reference	No of Patients	Place of Stimulation	Stimulation Frequency (Hz) and Train Duration (s)	Single Stimulus Duration and Strength (mA), Interval between Trains of Stimuli	Observation Time (Months), Numbers of Session	Evaluation Method of Effectiveness	Additional Comments
**Baldi** **et al.,** **1998 [25]**	26	hip extensors, knee extensors and knee flexors	60 Hz	0.375 ms 140 mA	6 months 3 times per week 30 min per day	Total body lean body mass, lower limb lean body mass, and gluteal lean body mass assessed by using dual-energy X-ray absorptiometry	All the parameters were fixed and the same for all the patients
**Kern** **et al.,** **2010 [27]**	22	leg muscles	-	12–150 ms 250 mA	2 years 5 times a week 30 min per day	EMG, knee torque measurement every 12 weeks, muscle cross-sectional area measured by computed tomography, muscle biopsies	Home-based therapy
**Pieber** **et al.,** **2015 [30]**	24	tibialis anterior muscle	1 Hz	200 ms 500 ms 8 mA	4 sessions, 1–2 min each	Medical Research Council Scale Intensity was recorded in absolute values in mA and relatively provided in percentage to the first stimulation of each patient.	-
**Lee** **et al.,** **2015 [29]**	22	median and common peroneal nerves	100 Hz 2 s	0.4 ms 12–30 mA	6 weeks, 5 days a week 30 min per day	EMG ENG non-stimulated leg used as a control	All the parameters were fixed and the same for all the patients
**Zheng** **et al.,** **2020 [17]**	5	knee extensors	30 Hz	30–99 mA	12 weeks of therapy, sessions twice a week	Quadriceps femoris torque, blood lipid profilin, c-reactive protein, spasticity evaluation, quality of life index	-
**Huber** **et al.,** **2021 [16]**	42	peroneal and tibial nerves	20–70 Hz 2–3 s	11–22 ms 18–45 mA 2–3 s	6–14 months, 5 days a week, 15–20 min per day	EMG ENG Results were compared to patients treated with kinesiotherapy alone	Home-based therapy Stimulus strength was the only parameter adjusted by patients

## Data Availability

All the data generated or analyzed during this study are included in this published article.

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
