# Peer review of "The Role of Transcranial Magnetic Stimulation, Peripheral Electrotherapy, and Neurophysiology Tests for Managing Incomplete Spinal Cord Injury"

_biomedicines, 2023, doi:10.3390/biomedicines11041035_

Round 1
Reviewer 1 Report
Dear Authors,
The manuscript looks like a report on scientific experiments. The experiments are an electrophysiological study of subjects with incomplete spinal cord injury before and after treatment. There are data on the demographic, anthropometric, and medical characteristics of patients. The patients were divided into three groups. One group underwent a course of electrotherapy of the peroneal and tibial nerves in combination with kinesiotherapy. Patients of the second group underwent transcranial magnetic stimulation of the motor cortex in combination with kinesiotherapy. Patients of the third group were treated only with kinesiotherapy. The researchers controlled the amplitude and frequency of the EMG of the anterior tibial muscle during maximum voluntary muscle contraction. This testing is done before and after the course. The manuscript looks like an experimental article.
At the same time, in the Methods of the manuscript, there is a paragraph "Literature review". An analysis was made of articles presenting data on the effect of transcranial magnetic stimulation on the motor rehabilitation of patients with spinal cord injuries. Similarly, the authors analyzed articles on electrical stimulation of motor nerves in patients with spinal cord injuries. Paragraphs 2.3 and 2.4 of the Methodology (“Transcranial magnetic stimulation (TMS)” and “Peripheral electrotherapy”, respectively) look like a brief review of the literature.
The authors refer to the articles " The Role of Peripheral Nerve Electrotherapy in Functional Recovery of Muscle Motor Units in Patients after Incomplete Spinal Cord Injury " (ref. 15) and " The Long-Term Effect of Treatment Using the Transcranial Magnetic Stimulation rTMS in Patients after Incomplete Cervical or Thoracic Spinal Cord Injury” (ref. 14). The authors of the manuscript are co-authors of these articles. The patient cohorts from these articles are the same as the patient cohorts from the manuscript. The rehabilitation protocols were the same in the articles and the manuscript. In all cases, before and after treatment, EMG of the anterior tibial muscle was studied at an arbitrary maximum contraction. Data analysis results differ, but not fundamentally, between the manuscript and articles.
So why does the manuscript duplicate data from articles?
What is this type of manuscript, experimental article, or literature review?
Major Comments
Why was tibialis anterior activity chosen for neurophysiological feedback? Why was the activity of other muscles of the lower extremities not recorded and analyzed?
The result of a single electrophysiological test is not a reliable argument when discussing the effectiveness of technologies for the treatment of spinal cord injury.
Please discuss why brain stimulation was less effective on tibialis anterior activity than transcranial magnetic stimulation. Perhaps the reason is that the tibial nerve was stimulated during the course. Perhaps when testing the activity of the thigh muscles, the difference will not be revealed.
Figure 1, E-G. Please show the entire period during which the subjects contracted the muscles (5 s) instead of the short period shown (~0.5 s).
Minor Comments
Please provide references to the standard kinesiotherapy algorithms mentioned on line 168.
Line 172. “b – sEMG recording …”. Please correct “c – sEMG recording …”
Line 176. “from the magnetic stimulation device “ms””. The “ms” absent in the figure.
“Table 4. Data on parameters of transcranial magnetic stimulation in patients with spinal cord injury gathered from available literature and from observations where effects confirmed its efficiency” should be Table 2. “Table 5. Data on parameters of peripheral electrostimulation in patients with spinal cord injury gathered from available literature and from observations where effects confirmed its efficiency” should be Table 3. These tables are mentioned in the manuscript before the data presented in subsequent tables.
Lines 351-362. The phrase doubles the phrase above (lines 357-359).
Line 365-367. “In all the groups of patients treated with two types of combined therapy or kinesiotherapy alone, there was no significant improvement in the parameters of the amplitude and frequency of sEMG recording from the tibialis anterior muscle during the attempt of maximal contraction”. This phrase contradicts the text in lines 352-363.
Lines 368, 369. “There were statistically comparable to those recorded in healthy volunteers from the control group”. The differences between data recorded in patients and data recorded in healthy individuals are evident when looking at Figure 1, E-G. Is this sentence correct?
Line 422. “Many of these algorithms pose a risk of iatrogenic nerve damage”. Please give a reference.
Best regards
Reviewer 2 Report
The article by Leszczyńska et al. "The role of transcranial magnetic stimulation, peripheral electrotherapy and neurophysiology tests for managing incomplete spinal cord injury" covers a potentially interesting and emerging topic related to the spinal cord injury therapy. In this sense, this remains to be potentially interesting for the Biomedicines readers.
I regard the main point of this paper as highly attractive as well as the results are clearly presented. The text does not contain any major errors, therefore I have some minor comments and recommendations:
1. There is a need to provide slightly more expanded introduction shortly
mentioning/describing pathogenesis of SCI and its impact of modern healthcare.
2.. The figure (diagramm) summarizing and clarifying the Literature review search should be added.
3. Following references should be added and properly cited within the main text:
- Kubaszewski Ł, Wojdasiewicz P, Rożek M, Słowińska IE, Romanowska-Próchnicka K, Słowiński R, Poniatowski ŁA, Gasik R. Syndromes with chronic non-bacterial osteomyelitis in the spine. Reumatologia. 2015;53(6):328-36. doi: 10.5114/reum.2015.57639.
- Brihmat N, Allexandre D, Saleh S, Zhong J, Yue GH, Forrest GF. Stimulation Parameters Used During Repetitive Transcranial Magnetic Stimulation for Motor Recovery and Corticospinal Excitability Modulation in SCI: A Scoping Review. Front Hum Neurosci. 2022 Apr 7;16:800349. doi: 10.3389/fnhum.2022.800349.
- Tykocki T, Poniatowski ŁA, Czyz M, Wynne-Jones G. Oblique corpectomy in the cervical spine. Spinal Cord. 2018 May;56(5):426-435. doi: 10.1038/s41393-017-0008-4.
- De Miguel-Rubio A, Muñoz-Pérez L, Alba-Rueda A, Arias-Avila M, Rodrigues-de-Souza DP. A Therapeutic Approach Using the Combined Application of Virtual Reality with Robotics for the Treatment of Patients with Spinal Cord Injury: A Systematic Review. Int J Environ Res Public Health. 2022 Jul 19;19(14):8772. doi: 10.3390/ijerph19148772.
4. In some places the use of English could be improved on.
Completing this gaps will have an impact on the understanding the aim of the study and, from my point of view, is absolutely necessary.
Round 2
Reviewer 1 Report
Dear Authors,
Thank you very much for your detailed answer to my review and manuscript revision.
I am sorry for the typo in the comment “Please discuss why brain stimulation was less effective on tibialis anterior activity than transcranial magnetic stimulation”. It should be “Please discuss why brain stimulation was less effective on tibialis anterior activity than electrical stimulation”. Your answer to this comment satisfied me.
I am satisfied with almost all of your comments and improvements to the article.
References which you added to the sentences “Many of these algorithms pose a risk of iatrogenic nerve damage” (lines 440, 441) are wrong. You referenced [27-29]. Ref. 28 is [Günter, C.; Delbeke, J.; Ortiz-Catalan, M. Correction to: Safety of Long-Term Electrical Peripheral Nerve Stimulation: Review of the State of the Art. J NeuroEngineering Rehabil 2020, 17, 77]. Probably it was a mistake and you planned to refer to [Günter, C., Delbeke, J., & Ortiz-Catalan, M. (2019). Safety of long-term electrical peripheral nerve stimulation: review of the state of the art. Journal of neuroengineering and rehabilitation, 16, 1-16]. Refs. 27 and 29 are the original articles on the effect of the course of peripheral nerve stimulation in spinal patients. Nice effects of the course were obtained in these studies and nothing side effects of the stimulation appeared. Please correct the references after “Many of these algorithms pose a risk of iatrogenic nerve damage” (lines 440, 441).
After you add two experimental groups it is an obvious lack of information about treatment duration. The rTMS course duration is described in lines 240-243. The time of other interventions is not presented in the manuscript. The same important information about the regimen of the combination of muscle stimulation or brain stimulation and kinesitherapy. These data are extremely important to everybody reader who will plan to repeat your results. Please add this information to the Methods.
Best regards,
Reviewer
